# Combining sCD163 with CA 19-9 Increases the Predictiveness of Pancreatic Ductal Adenocarcinoma

**DOI:** 10.3390/cancers15030897

**Published:** 2023-01-31

**Authors:** Liva K. Stuhr, Kasper Madsen, Astrid Z. Johansen, Inna M. Chen, Carsten P. Hansen, Lars H. Jensen, Torben F. Hansen, Kirstine Kløve-Mogensen, Kaspar R. Nielsen, Julia S. Johansen

**Affiliations:** 1Department of Oncology, Copenhagen University Hospital-Herlev and Gentofte, DK-2730 Herlev, Denmark; 2Department of Surgery, Copenhagen University Hospital-Rigshospitalet, DK-2200 Copenhagen, Denmark; 3Department of Oncology, University Hospital of Southern Denmark, DK-7100 Vejle, Denmark; 4Department of Clinical Immunology, Aalborg University Hospital, DK-9000 Aalborg, Denmark; 5Department of Medicine, Copenhagen University Hospital-Herlev and Gentofte Hospital, DK-2730 Herlev, Denmark; 6Department of Clinical Medicine, Faculty of Health and Medical Sciences, University of Copenhagen, DK-2200 Copenhagen, Denmark

**Keywords:** biomarker, pancreatic cancer, soluble CD163, tumor-associated macrophages

## Abstract

**Simple Summary:**

During the last decades, the CA 19-9 blood test has been the only widely used biomarker in patients with pancreatic ductal adenocarcinoma (PDAC). Given the poor prognosis and staggering mortality rates of this type of cancer, partly due to late diagnosis, new and easily available biomarkers are in high demand. Using a large cohort of patients with PDAC, we found that a combination of CA 19-9 and sCD163 blood tests was a superior diagnostic marker compared to the recommended CA 19-9 test alone. Our findings suggest that sCD163 could have clinical value as a novel, minimally invasive, and cost-effective diagnostic marker. However, because this is the first study examining sCD163 in patients with PDAC, further studies are needed to validate our findings.

**Abstract:**

The objective of this study was to evaluate the diagnostic and prognostic potential of soluble CD163 (sCD163) in patients with pancreatic ductal adenocarcinoma (PDAC). Preoperative serum samples from 255 patients with PDAC were analyzed for sCD163 using a commercially available enzyme-linked immunosorbent assay. The diagnostic value of sCD163 was evaluated using receiver operating characteristic (ROC) curves. The prognostic significance of sCD163 was evaluated by Cox regression analysis and Kaplan–Meier survival curves. sCD163 was significantly increased in patients with PDAC, across all stages, compared to healthy subjects (stage 1: *p* value = 0.033; stage 2–4: *p* value ≤ 0.0001). ROC curves showed that sCD163 combined with CA 19-9 had the highest diagnostic potential compared to sCD163 and CA 19-9 alone both in patients with local PDAC and patients with advanced PDAC. Univariate and multivariate analysis showed no association between sCD163 and overall survival. This study found elevated levels of circulating sCD163 in patients with PDAC, regardless of stage, compared to healthy subjects. This suggests that sCD163 may have a clinical value as a novel diagnostic biomarker in PDAC.

## 1. Introduction

Pancreatic ductal adenocarcinoma (PDAC) is an aggressive malignancy with a high mortality rate and a 5-year survival rate of around 10% [1,2]. The incidence is increasing worldwide in step with the aging population, and PDAC is expected to become the second-leading cause of cancer-related mortality by 2030 [3]. At the time of diagnosis, less than 25% of patients have a resectable tumor, and following radical resection, the average 5-year survival is around 20% [4].

The diagnostic and therapeutic advances made during the last decades have only had a modest impact on patient outcomes. The complex heterogeneity of PDAC is one reason for the lack of prognostic or predictive biomarkers. Carbohydrate antigen 19-9 (CA 19-9) in plasma is the most widely studied biomarker to predict survival in patients with metastatic PDAC [5]. The presence of a systemic inflammatory response as measured by C-reactive protein (CRP), interleukin-6 (IL-6), and YKL-40 (also termed chitinase 3-like 1 protein [CHI3L1]) appears to be a useful indicator of a poor prognosis in patients with PDAC [6].

The poor prognosis of PDAC is due to late diagnosis and resistance to standard chemotherapy. This is partly due to the complex and highly immunosuppressive tumor microenvironment (TME) that inhibits the anti-tumor immune responses and promotes the growth, angiogenesis, invasion, and metastasis of the cancer cells [7]. Of particular interest is the tumor-associated macrophage (TAM), a specific type of inflammatory cell in the TME [4]. TAMs can be divided into two general subtypes: classically activated M1 macrophages, which are associated with pro-inflammatory properties, and alternatively activated M2 macrophages, which are associated with anti-inflammatory properties. In PDAC, the presence of M2-polarized TAMs is associated with a poor prognosis [8].

CD163 is a hemoglobin scavenger receptor on monocytes and macrophages. CD163 is related to inflammation and is considered a marker of alternatively activated M2 macrophages [9]. M2-polarized macrophages that express CD163 promote angiogenesis and the production of matrix metalloproteinases, which enhances tumor growth and invasion [10]. Increased infiltration of CD163-expressing cells in the highly desmoplastic TME of PDAC has been associated with shorter overall survival (OS) [11]. Several studies have found that increased tumor infiltration by CD163-expressing macrophages is associated with shorter OS in patients with PDAC [12,13,14,15] (Appendix A). This is consistent with studies of other types of cancer that have found that increased infiltration of CD163-expressing cells is associated with shorter OS in follicular lymphoma [16], triple-negative breast cancer [17], and hepatocellular carcinoma [18].

Soluble CD163 (sCD163) is released into plasma as a result of ectodomain shredding with proteolysis of membrane proteins [19]. Elevated circulating sCD163 levels are found in patients with different types of cancer and can be used to estimate the total-body M2 macrophage load [9]. Elevated sCD163 levels have been associated with poor survival in patients with different types of cancer, including multiple myeloma [20], gastric cancer [21], colorectal cancer [22], melanoma [23], hepatocellular carcinoma [24,25], epithelial ovarian cancer [26], classical Hodgkin’s lymphoma [27], B-cell lymphocytic leukemia [28] and diffuse large B-cell lymphoma [29] but not in patients with renal cell carcinoma [30] (Appendix A). A recent systematic review and meta-analysis including 8 papers and 1236 patients with cancer concluded that a higher circulating level of sCD163 was significantly correlated with shorter OS and progression-free survival in patients with different types of cancer [31]. No studies have reported if circulating sCD163 has prognostic value in patients with PDAC.

The aim of this biomarker study was to test the hypotheses that circulating sCD163 levels are elevated in patients with PDAC compared to healthy subjects and that high plasma sCD163 is associated with poor prognosis in patients with PDAC.

## 2. Materials and Methods

### 2.1. Patients

We analyzed pretreatment serum samples from 255 patients included in the Danish BIOPAC study “BIOmarkers in patients with PAncreatic Cancer (BIOPAC)—can they provide new information of the disease and improve diagnosis and prognosis of the patients?” (ClinicalTrials.gov ID: NCT03311776; www.herlevhospital.dk/BIOPAC/, accessed on 1 December 2022). The BIOPAC study is a prospective multicenter open cohort study with ongoing enrollment. Biological samples and clinical data are collected prospectively in patients with localized, locally advanced, or metastatic pancreatic tumors treated at seven Danish hospitals. Blood samples used in the present study were collected from patients included at Copenhagen University Hospitals at Herlev and Rigshospitalet and Vejle Hospital). The patients included in the BIOPAC study are followed from the time of diagnosis and during treatment and follow-up until death.

The patients received oral and written information prior to their enrollment and gave their written consent at baseline according to the guidelines of the Danish Ethics Committee. The BIOPAC study protocol has been approved by the Danish Ethics Committee (VEK, j.nr. KA-20060113) and the Danish Data Protection Agency (j.nr. 2006-41-6848, 2012-58-0004; HGH-2015-027; I-Suite j.nr. 03960; and PACTICUS P-2020-834) [32]. The study was conducted in accordance with the Declaration of Helsinki.

A total of 255 patients enrolled in the BIOPAC from 12 January 2012 to 3 April 2020 were included in the retrospective biomarker study of plasma sCD163. The patients were followed until 9 April 2022 or death. The surgical and oncologic treatment was given in accordance with Danish national guidelines (http://www.gicancer.dk/, accessed on 1 December 2022) (Appendix A).

The following baseline characteristics were selected from the BIOPAC database: date of inclusion in the BIOPAC study, date of death, age, performance status (PS), diabetes, Charlson comorbidity index (CCI), cachexia, smoking status and alcohol intake, presence of intrabiliary stent, PDAC stage, tumor location and size, metastases and the number of metastatic sites.

A total of 80 healthy controls from Aalborg University Hospital were included as controls. The median age of the healthy controls was 66 years (range 36–69). The healthy controls were chosen based on age to match the patients with PDAC and thereby ensure a more accurate comparison.

### 2.2. Definition of Covariates

The CCI is a validated, simple, and readily applicable method of estimating the risk of death from comorbid disease. Higher CCI scores are associated with a greater mortality risk and comorbid disease severity. The age-adjusted Charlson comorbidity index (CACI) was calculated as the CCI after adding 1 point for every 10-year increase from 40 years of age [33,34].

The ECOG (Eastern Cooperative Oncology Group) scale of PS is used to quantify the functional status of cancer patients and is an important factor in determining prognosis [35].

Cachexia was defined as a weight loss of more than 5% over the past 6 months or as a BMI under 20 and any degree of weight loss of more than 2% [36].

Alcohol abuse was defined as alcohol consumption of more than 7 drinks per week for women and more than 14 drinks per week for men (1 drink ≈ 12 g of alcohol).

### 2.3. Biological Samples

The blood for biochemical analysis was drawn at the time of diagnosis and before surgery or the commencement of chemotherapy. The samples were processed according to the BIOPAC protocol. All blood samples were centrifuged at 2300× *g* at 4 °C for 10 min, and serum was aliquoted in Greiner tubes (Cryo.s™ Freezing Tubes, 2 mL, GR-121280, Greiner Bio-One GmbH, Baden-Württemberg, Germany). The serum was subsequently stored at −80 °C.

### 2.4. sCD163 ELISA

sCD163 in serum was determined using a solid phase sandwich enzyme-linked immunosorbent assay (ELISA) (Quantikine^®^ ELISA, R&D Systems, Abingdon, UK) in accordance with the manufacturer’s instructions. Here, 96-well microplates were precoated with a monoclonal antibody specific for human sCD163, and 100 μL of assay diluent (RD1-34, buffered protein base) was added to each well. Subsequently, 50 μL of each sample (per plate: 8 standards, 3 assay controls, and 37 serum samples (diluted 1:40) from patients and healthy controls) were added per well before incubation for 2 h at room temperature. The plate was washed four times with wash buffer (25-fold concentrated solution of buffered surfactant). After washing, 200 μL of human CD163 conjugate (monoclonal antibody specific for human CD163 conjugated to horseradish peroxidase) was added, followed by 2 h of incubation at room temperature. The plate was washed four times using the wash buffer, and 200 μL of substrate solution (color reagent A, stabilized hydrogen peroxide, and color reagent B, stabilized chromogen) was added. The plates were incubated for 30 min at room temperature. Finally, the reaction was stopped by adding 50 μL of the stop solution (2 N sulfuric acid). The optical density was determined using a microplate reader set to a wavelength of 450 nm. A standard curve was constructed by plotting the mean absorbance for each standard on the *y*-axis against the concentration of the standards on the *x*-axis. A second-order polynomial equation was used to fit the standards, and quantitative sCD163 concentrations were determined by comparison of the optical density values with the standard curve. The manufacturer reports an intra-assay coefficient of variation (CV) (20 samples) for sCD163 to be 3.8% (low control = 20 ng/mL), 3.4% (medium control = 35 ng/mL), and 3.5% (high control = 66 ng/mL). An inter-assay CV (40 assays) was found to be 6.7% (low control = 20 ng/mL), 4.6% (medium control = 35 ng/mL), and 4.1% (high control = 64 ng/mL).

### 2.5. CA 19-9, CRP, IL-6, and YKL-40 Assays

Concentrations of CA 19-9 were measured using the IMMULITE 2000 GI-MA assay (Siemens, Catalogue Number L2KG12), which is a solid-phase, two-site sequential chemiluminescent immunometric assay. Imprecision was monitored with two internal controls at 16 and 83 U/mL with coefficients of variation of 8% and 9%. Accuracy was monitored within the standard UK NEQAS program. Elevated CA 19-9 was defined as >37 U/mL.

High-sensitive CRP was measured using a sensitive CRP ultra ready-to-use liquid assay reagent by an immunoturbidimetric method on a fully automated chemistry analyzer (Kit-test SENTINEL CRP Ultra (UD), 11508 UD-2.0/02 2015/09/23). The measurement range was 0.3–640 mg/L. The intra- and inter-assay CVs were 3% and <15%.

Plasma IL-6 was measured using a high-sensitive ELISA (Quantikine HS600B, R&D Systems, Abingdon, UK) in accordance with the manufacturer’s instructions. The lower limit of detection for IL-6 was 0.01 ng/L, and the intra- and inter-assay CVs were ≤8% and ≤11%.

Plasma YKL-40 was measured using an ELISA (Quidel Corporation, San Diego, CA, USA) in accordance with the manufacturer’s instructions. The lower limit of detection for YKL-40 was 20 μg/L, and the intra- and inter-assay CVs were <5% and <6%.

### 2.6. Statistical Analysis

Results are reported in accordance with the REMARK (reporting recommendations for tumor marker prognostic study) guidelines [37] (Appendix A). This is an exploratory study, and no sample size was calculated.

The concentration of plasma sCD163 was described by the median and range. The Wilcoxon signed-rank test was used to assess potential differences in plasma sCD163 concentrations across categorical groups consisting of two subgroups. For categorical variables of more than two subgroups, the Kruskal–Wallis test was used.

Pairwise Wilcoxon signed-rank tests were conducted for measurements grouped by cancer stage, including measurements from healthy controls (Appendix A). These results were corrected for multiple comparisons by the Holm step-down procedure and presented in a boxplot.

Potential monotonic relationships between plasma sCD163 and selected biomarkers (CA 19-9, CRP, IL-6, and YKL-40) were analyzed using the Spearman rank correlation coefficient.

Receiver operating characteristic (ROC) analysis was performed to access the diagnostic value of sCD163 and CA 19-9. A logistic regression model was used to combine the results from plasma levels of sCD163 and CA 19-9 to enhance the accuracy.

For survival analysis, the population was divided into two groups based on the cancer stage and analyzed individually. Patients were grouped into three groups based on the tertiles of the observed sCD163 concentrations. Survival curves were produced using the Kaplan–Meier method. Cox regression was conducted in three steps. First, univariable Cox regression was conducted to assess the relationship between sCD163 (log-transformed) and OS. This was also carried out for a selected group of potential confounders like CA 19-9, CRP, IL-6, and YKL-40 concentrations (log-transformed), PS, age, and sex. Then multivariable Cox regression with sCD163 adjusted for age and PS was performed. Last, a multivariable Cox regression with sCD163 adjusted for all the previously selected potential confounders was performed.

## 3. Results

### 3.1. Characteristics of Study Population

Cohort characteristics are summarized in Table 1. A total of 100 samples were obtained from patients with resected localized PDAC, and 155 samples were obtained from patients with non-resectable tumors due to locally advanced or metastatic disease. Patients aged 70 years and over accounted for over one-third of the included patients. The majority of patients (60%) had metastatic disease at the time of diagnosis and sample collection. Cachexia was registered in more than half of the included patients. In total, 88 patients had biliary tract stents.

### 3.2. sCD163 According to Clinical and Tumor Characteristics

sCD163 concentrations in relation to clinical and tumor characteristics are shown in Table 1. Older patients and patients with cachexia or a stent had higher sCD163. There were no differences in sCD163 in relation to PS, diabetes, stage, CACI, BMI, smoking and drinking status, tumor size, tumor location, and metastatic sites and number.

Patients with stage 2 (*n* = 48), 3 (*n* = 42), and 4 (*n* = 153) had higher sCD163 concentrations compared to healthy controls (*p* < 0.0001) (Figure 1A and Appendix A). Patients with stage 1 (*n* = 12) had slightly higher sCD163 concentrations than healthy controls (*p* = 0.033) (Figure 1A). No difference was found in sCD163 in resected patients according to tumor size (Figure 1B). Patients with localized PDAC and tumor location in the caput had significantly higher sCD163 levels compared to healthy controls (*p* < 0.0001) (Figure 1C). Females in stage 1 PDAC had higher sCD163 compared to men in stage 1 (Appendix A). No difference in sCD163 was found in patients with PDAC according to diabetes, BMI, and cachexia when the patients were stratified according to localized or advanced disease (Appendix A).

Low correlations were found between sCD163 and bilirubin, IL-6, YKL-40, alkaline phosphatase (ALP), alanine aminotransferase (ALAT), and platelets and no correlation with CA 19-9, CRP, and aspartate aminotransferase (ASAT) (Table 2).

ROC analysis was conducted to access the diagnostic capacity of sCD163 and CA 19-9. In all patients, CA 19-9 (AUC = 0.91, 95% CI 0.88–0.95) was superior to sCD163 (AUC = 0.78, 0.73–0.83). The combined prediction of sCD163 and CA 19-9 showed the highest diagnostic potential (AUC = 0.95, 0.93–0.98) (Figure 2A). Similar results were found in the resected patients (AUC = 0.93, Figure 2B) and in patients with advanced PDAC (AUC = 0.98, Figure 2C).

### 3.3. sCD163 and Overall Survival

Kaplan–Meier survival curves in the present cohort of patients with PDAC according to stage are shown in Appendix A and confirm the poor survival in patients with stage 4. The Kaplan–Meier survival curves according to sCD163 (divided in tertiles) in patients with stage 1 + 2 (Figure 3A) and stage 3 + 4 (Figure 3B) are shown in Figure 3. High sCD163 concentrations were not associated with poor survival in patients with PDAC.

Univariate Cox regression analysis in patients with stage 1 + 2 PDAC showed that sCD163 was not a prognostic biomarker (Table 3). Nor did uni- and multivariate Cox regression analysis (sCD163, CRP, CA 19-9, IL-6, YKL-40, PS, age, and sex) in patients with stage 3 + 4 PDAC show that sCD163 was a prognostic biomarker (Table 4). In these patients with advanced PDAC, CRP, CA 19-9, YKL-40, PS, and age were independent prognostic variables.

## 4. Discussion

This is the first large-scale study to investigate circulating sCD163 in samples from a population-based consecutive cohort of patients with PDAC. We found that sCD163 was elevated in patients with PDAC (all stages) compared to healthy subjects. This confirms our hypothesis that circulating sCD163 levels are elevated in patients with PDAC and indicates that sCD163 may have diagnostic value in patients with PDAC. Our results agree with previous studies of various malignancies, where sCD163 was significantly higher in patients with cancer compared to healthy subjects [21,24,29,30].

The mechanisms that contribute to the increased circulating sCD163 in patients with PDAC remain elusive. Macrophages are broadly recognized to mediate inflammatory reactions and immune responses, as well as to facilitate cancer invasion, migration, and subsequent metastasis [38]. CD163, which acts as a specific macrophage activation marker, is commonly up-regulated when macrophages are stimulated by IL-6, IL-10, glucocorticoid, and macrophage colony-stimulating factor [19]. Inflammatory stimulation, activation of Toll-like receptors, and participation of lipopolysaccharide (LPS), metalloprotease and inflammatory medium could collectively contribute to the shredding of CD163 from the surface membrane of the macrophage and into plasma in a soluble form [39]. The circulating level of sCD163 is therefore considered to reflect the degree of the local inflammatory response and abundance of M2-polarized macrophages associated with cancer-promoting functions in the TME. It should be noted that there is a certain similarity between the biological behavior of anti-inflammation immunity and anticancer immunity. The elevated levels of circulating sCD163 may thus be partially ascribed to the activation of macrophages during the process of anticancer immunity.

An important finding of our study was that sCD163 was significantly elevated regardless of cancer stage. This suggests that sCD163 may be a novel biomarker for PDAC. We evaluated the diagnostic ability of sCD163 and compared it to that of CA 19-9, which is the most widely used biomarker for PDAC. Notably, the diagnostic ability was enhanced when sCD163 and CA 19-9 were combined in patients with local PDAC before resection and in patients with advanced PDAC before palliative chemotherapy. Therefore, it should be further investigated whether circulating sCD163 in combination with CA 19-9 is a novel PDAC diagnostic biomarker that might even be superior to CA 19-9 alone if used in combination with other panels of diagnostic proteins [40,41,42].

As expected, low correlations were found between sCD163 and inflammatory biomarkers (CRP, IL-6, YKL-40, and platelets), but we found no association between sCD163 and diabetes, BMI, and cachexia when the patients were stratified according to localized or advanced PDAC. sCD163 was also correlated in all patients with bilirubin and in the operated patients highest levels of sCD163 were found in patients with a tumor localized in the caput of the pancreas.

Furthermore, we investigated potential associations between plasma sCD163 and adverse outcomes in patients with PDAC. Higher sCD163 concentrations at the time of diagnosis showed no association with poor survival in patients with PDAC. This suggests that sCD163 does not have the significant prognostic value that we hypothesized it did. To the best of our knowledge, no previous study has evaluated the association between circulating sCD163 and survival in patients with PDAC. Our findings are thus in contrast with those of previous studies in patients with other types of cancer that have reported a significant association between increased circulating levels of sCD163 and shorter survival [20,21,22,23,24,25,26,27,28,29].

PDAC is characterized by the presence of a particularly prominent desmoplastic stroma that accounts for the majority of the tumor volume [43]. The stromal component is believed to play an important role in limiting the vascularization of the tumor as well as recruiting immune cells. Given the previously reported association between increased tumor infiltration by CD163 macrophages with shorter survival, the abundant amount of stroma and resulting limitations may explain why we did not find a correlation between circulating sCD163 and prognosis [12,13,14,15]. A diagnostic biomarker will not necessarily have potential as a prognostic biomarker. This is illustrated in two recent studies by Lindgaard S et al. that investigated the diagnostic and prognostic potential of 92 different proteins in patients with PDAC. Only two of these proteins seemed to have both diagnostic and prognostic potential [40,44].

In the future, we plan to analyze circulating sCD163 in patients with PDAC during treatment with palliative chemotherapy and immunotherapy. In some of these patients, we will also study the relationship between circulating sCD163 and CD163 expression in tumor tissue.

There are several strengths and limitations of this study. The strengths of this study include its prospective design, which minimizes potential sources of bias and confounding. Another strength is the large study cohort of patients with PDAC and the relatively long follow-up period. In addition, the study also has a large group of age-corresponding healthy controls. An important limitation of the study is that our results have not yet been confirmed in a validation cohort. A separate study performed within different clinical trial data sets is required to increase the level of evidence that supports the use of our tumor biomarker test [45]. Our knowledge is limited about the human CD163 Quantikine^®^ ELISA that we used to analyze our samples. The company (R&D Systems) has not stated whether a method study regarding this specific marker has been conducted. Finally, it is unknown to what extent sCD163 correlates with M2-polarized macrophage behavior in the TME as tumor biopsies were not available in this study.

## 5. Conclusions

In conclusion, this is the first study to evaluate the diagnostic and prognostic value of sCD163 in patients with PDAC. Our study found elevated levels of circulating sCD163 in patients with PDAC compared to healthy subjects. However, we did not find any evidence to support the use of sCD163 as a prognostic biomarker because there was no association between sCD163 and OS. Our findings suggest that sCD163 may have clinical value as a novel, minimally invasive, and cost-effective diagnostic marker. Future studies are needed to further evaluate both the prognostic and diagnostic value of plasma sCD163 in patients with PDAC.

## Figures and Tables

**Figure 1 cancers-15-00897-f001:**
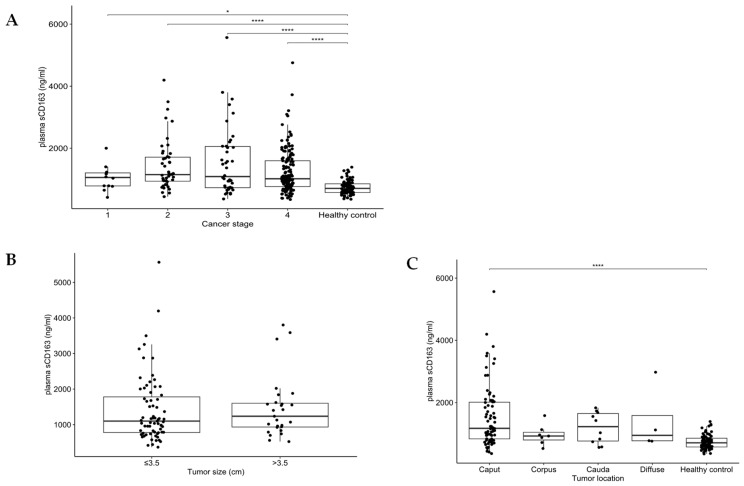
Box plot showing the distribution of sCD163 concentrations in patients with pancreatic ductal adenocarcinoma at time of diagnosis stratified by stage, and in healthy controls (**A**), by tumor size in resected patients (**B**), and by tumor location in resected patients (**C**). * = *p* value = 0.033; **** = *p* value ≤ 0.0001.

**Figure 2 cancers-15-00897-f002:**
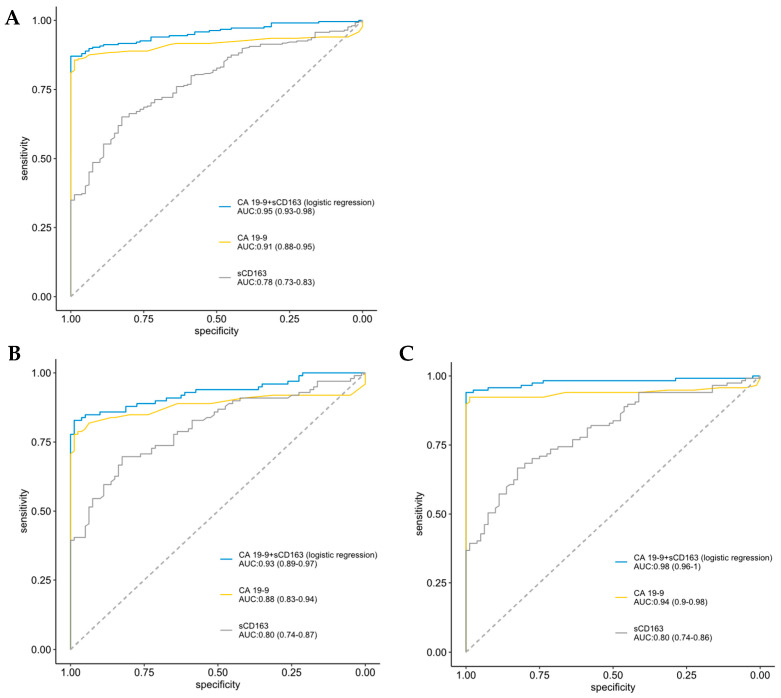
ROC curves for single sCD163, single CA19-9 and the combination of them in discriminating all patients (**A**), resected patients (**B**), and patients with locally advanced and metastatic pancreatic ductal adenocarcinoma (**C**) from healthy controls.

**Figure 3 cancers-15-00897-f003:**
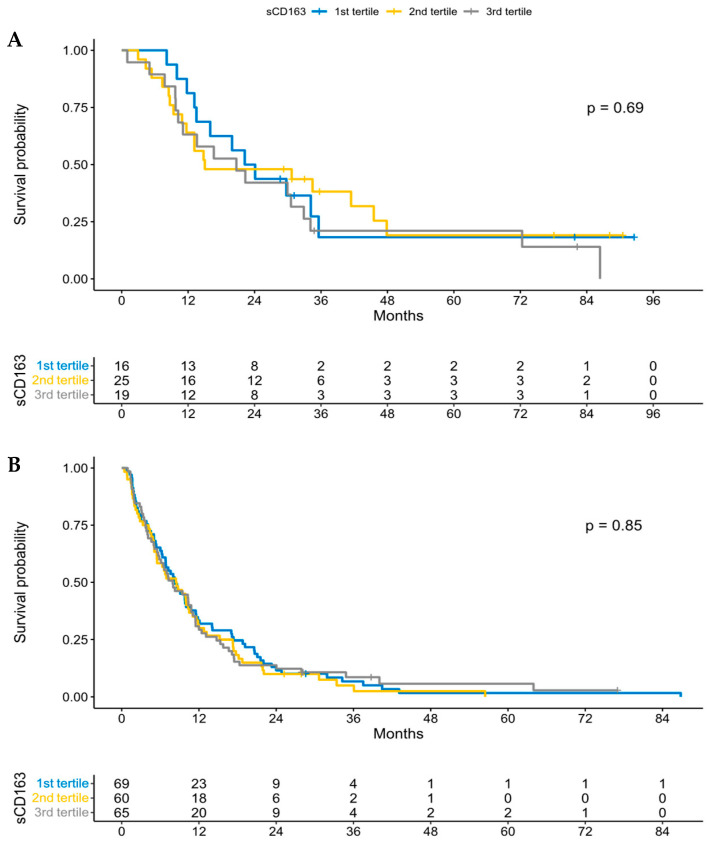
Kaplan–Meier survival curve showing overall survival according to sCD163 concentrations at time of diagnosis, divided into tertiles, (**A**) in patients with stage 1 + 2 pancreatic ductal adenocarcinoma and (**B**) stage 3 + 4 pancreatic ductal adenocarcinoma.

**Table 1 cancers-15-00897-t001:** Pretreatment sCD163 levels stratified according to clinical and tumor characteristics of the 255 patients with pancreatic ductal adenocarcinoma.

Pancreatic Ductal Adenocarcinoma	N = 255 (%)	sCD163 Median (Range)	*p*-Value
Age			0.013
<50 years	14 (5.5)	886 (441–3407)	
50–70 years	140 (54.9)	1046 (350–4196)	
>70 years	101 (39.6)	1135 (368–5568)	
Sex			0.018
Male	134 (52.5)	982 (368–3587)	
Female	121 (47.5)	1157 (350–5568)	
Performance status			0.287
0	92 (36.1)	1108 (441–4196)	
1	146 (57.3)	1024 (350–5568)	
≥2	17 (6.7)	1062 (466–2763)	
Diabetes			0.800
Yes	70 (27.5)	1059 (350–3802)	
No	185 (72.5)	1058 (394–5568)	
CACI			0.062
0–1	31 (12.2)	834 (441–3407)	
2–3	119 (46.7)	1126 (419–4196)	
≥4	69 (27.1)	1135 (350–5568)	
BMI			0.790
<18.5	16 (6.4)	1036 (394–2872)	
18.5–25	135 (54.2)	1060 (368–5568)	
>25	98 (39.4)	1071 (350–4756)	
Cachexia			0.042
Yes	135 (52.9)	1188 (368–5568)	
No	73 (28.6)	1037 (350–4196)	
Smoking status			0.243
Currently/Previously	159 (62.4)	1017 (368–4756)	
No	91 (35.7)	1084 (350–5568)	
Alcohol status			0.618
Abuse/Previous abuse	60 (23.5)	1060 (466–3802)	
No abuse	189 (74.1)	1057 (350–5568)	
Stent			0.010
Yes	88 (34.5)	1196 (441–5568)	
No	167 (65.5)	1011 (350–4756)	
Stage			0.319
1 + 2	60	1138 (419–4196)	
3 + 4	195	1042 (350–5568)	
Tumor size			0.808
>median (3.5 cm)	120 (47.1)	1061 (388–4756)	
≤median (3.5 cm)	122 (47.8)	1047 (350–5568)	
Tumor location			0.341
Caput	151 (59.2)	1080 (368–5568)	
Corpus	50 (19.6)	975 (350–3725)	
Cauda	42 (16.5)	1192 (531–3043)	
Diffuse	9 (3.5)	1120 (760–2975)	
Papillary	2 (0.8)	847 (637–1057)	
Metastatic sites			0.515
None	102 (40.0)	1126 (368–5568)	
Liver Only	81 (31.8)	1014 (350–4756)	
Liver + Lung	19 (7.5)	1105 (637–2528)	
Liver + Carcinosis	16 (6.3)	1006 (394–3094)	
Other	37 (14.5)	1018 (388–2763)	
Number of metastatic sites			0.352
0	102 (40.0)	1126 (368–5568)	
1	99 (38.8)	1042 (350–4756)	
≥2	54 (21.2)	1014 (388–3094)	

Missing values: BMI *n* = 6; CACI *n* = 36; cachexia *n* = 47; smoking status *n* = 5; alcohol status *n* = 6; tumor size *n* = 13; tumor location *n* = 1.

**Table 2 cancers-15-00897-t002:** Correlations between sCD163 and CA19-9, YKL-40, IL-6, CRP, ALP, bilirubin, ALAT, platelets, leucocytes, and neutrophils.

	Number of Observations	sCD163 Spearman’s ρ (95% CI)	*p*-Value
CA19-9	216	0.03 (−0.10–0.17)	0.616
YKL-40	253	0.23 (0.11–0.34)	0.0002
IL-6	253	0.20 (0.08–0.32)	0.001
CRP	183	0.11 (−0.04–0.25)	0.142
ALP	168	0.20 (0.05–0.34)	0.008
Bilirubin	188	0.28 (0.14–0.40)	0.0001
ALAT	183	0.23 (0.09–0.37)	0.002
ASAT	91	−0.03 (−0.23–0.18)	0.780
Platelets	189	0.17 (0.03–0.31)	0.019
Leucocytes	123	0.01 (−0.16–0.19)	0.875
Neutrophils	95	−0.01 (−0.21–0.20)	0.950

**Table 3 cancers-15-00897-t003:** Univariate Cox regression analyses for overall survival according to sCD163, CRP, CA19-9, IL-6 and YKL-40 concentrations (log-transformed), performance status, age, and sex, for patients with stage 1 + 2 pancreatic ductal adenocarcinoma.

Variable	Observations (Events)	HR (95% CI)	*p*-Value
sCD163	60 (47)	1.19 (0.81–1.76)	0.373
CRP	42 (32)	1.06 (0.90–1.24)	0.479
CA19-9	59 (46)	1.09 (0.99–1.20)	0.086
IL-6	59 (47)	1.07 (0.88–1.31)	0.490
YKL-40	59 (47)	1.20 (0.98–1.48)	0.080
PS	60 (47)		
0	26 (19)	Reference	
1	32 (26)	1.25 (0.69–2.26)	0.463
≥2	2 (2)	2.36 (0.54–10.30)	0.252
Age	60 (47)		
≤70 years	39 (31)	Reference	
>70 years	21 (16)	1.39 (0.74–2.59)	0.304
Sex			
Male	29 (24)	Reference	
Female	31 (23)	0.79 (0.45–1.41)	0.432

**Table 4 cancers-15-00897-t004:** Univariate Cox regression analyses for OS according to sCD163, CRP, CA19-9, IL-6 and YKL-40 concentrations (log-transformed), PS, age, and sex, for patients with stage 3 + 4 pancreatic ductal adenocarcinoma. #, numbers.

	Univariate Analysis			Multivariate Analysis		
Variable	Observations (Events)	HR (95% CI)	*p*-Value	Observations (Events)	HR (95% CI)	*p*-Value
sCD163	194 (187)	0.99 (0.83–1.19)	0.926	139 (133)	0.92 (0.72–1.18)	0.518
CRP	140 (134)	1.28 (1.17–1.41)	<0.0001	139 (133)	1.17 (1.04–1.31)	0.011
CA19-9	157 (150)	1.12 (1.07–1.18)	<0.0001	139 (133)	1.12 (1.06–1.19)	<0.0001
IL-6	194 (187)	1.17 (1.09–1.25)	<0.0001	139 (133)	1.01 (0.90–1.13)	0.863
YKL-40	194 (187)	1.42 (1.24–1.64)	<0.0001	139 (133)	1.22 (1.01–1.53)	0.037
PS	194 (187)			139 (133)		
0	66 (62)	Reference		56 (52)	Reference	
1	113 (111)	1.62 (1.18–2.22)	0.003	74 (73)	1.24 (0.85–1.81)	0.256
≥2	15 (14)	2.73 (1.51–4.92)	0.0009	9 (8)	2.63 (1.23–5.88)	0.013
Age	194 (187)			139 (133)		
≤70	115 (110)	Reference		85 (81)	Reference	
>70	79 (77)	1.74 (1.29–2.35)	0.0003	54 (52)	2.08 (1.39–3.10)	0.0004
Sex	194 (187)			139 (133)		
Male	104 (102)	Reference		72 (70)	Reference	
Female	90 (85)	1.02 (0.76–1.36)	0.909	67 (63)	1.094 (0.75–1.59)	0.639

## Data Availability

Data can be made available by contacting the last author.

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
