# Peer review of "Combining sCD163 with CA 19-9 Increases the Predictiveness of Pancreatic Ductal Adenocarcinoma"

_cancers, 2023, doi:10.3390/cancers15030897_

Round 1

Reviewer 1 Report

Overall, the study by Stuhr et al. is well written and evaluates an interesting research question on the potential diagnostic and prognostic capabilities of sCD163 for PDAC. I, however, do have a few questions and comments for the authors to address and consider:

1. Were serum or plasma samples used for the measurement of sCD163? The title and several parts of the manuscript notes plasma yet the methods section describes serum sample, as well as a couple of other sections in the results.

2. What was the % intra and inter CV for sCD163?

3. The result of no significant difference between diabetes is a bit surprising given studies having shown a drastic increase in sCD163 concentrations with impaired insulin sensitivity, beta-cell function, and incidence. Is there a potential explanation for this in this cohort?

 4. The AUC results show that CA 19-9 is quite a strong marker on its own and the addition of sCD163 only marginally improved prediction - not sure whether sCD163 presents any additional benefit in a clinical setting. Also, were these ROC/ AUC measurements also further adjusting for other clinical markers?

5. The follow-up duration of the study allows for some interesting associations to be investigated. Are there any additional samples at later time points to assess whether there are changes in sCD163? 

6. Being overweight or obese have been associated with increased sCD163 measurements and present increased risk for pancreatic cancer - yet this was not taken into account or presented in this paper. How does adjustment for waist circumference/ BMI affect the findings?

7. Not discussed in the manuscript are the results from Supplementary Figure S2 - i find the sex differences, particularly, in stage 1 quite interesting. How can you explain these findings and were the differences at each stay by sex significantly different?   

Author Response

The overall response to reviewer 1: We sincerely thank the reviewer for the positive evaluation of our manuscript. We also thank the reviewer for taking the time to thoroughly read the manuscript and for the constructive critique, ultimately improving the manuscript. Below is the point-by-point response to the comments.

Comments from Reviewer 1

Overall, the study by Stuhr et al. is well written and evaluates an interesting research question on the potential diagnostic and prognostic capabilities of sCD163 for PDAC. I, however, do have a few questions and comments for the authors to address and consider:

  1. Were serum or plasma samples used for the measurement of sCD163? The title and several parts of the manuscript notes plasma yet the methods section describes serum sample, as well as a couple of other sections in the results.

Author response: We used serum to measure sCD163, but we prefer only to describe this in the Method section since circulating sCD163 in the body is plasma. We have deleted some of the words “serum” or “plasma” in the text.

  1. What was the % intra and inter CV for sCD163?

Author response: The manufacturer (R&D Systems) reports an intra-assay coefficient of variation (CV) (20 samples) for sCD163 to be 3.8% (low control = 20 ng/ml), 3.4% (medium control = 35 ng/ml), and 3.5% (high control = 66 ng/ml). The inter-assay CV (40 assays) is reported to be 6.7% (low control = 20 ng/ml), 4.6% (medium control = 35 ng/ml), and 4.1% (high control = 64 ng/ml). The information has been added to the Method section.

  1. The result of no significant difference between diabetes is a bit surprising given studies having shown a drastic increase in sCD163 concentrations with impaired insulin sensitivity, beta-cell function, and incidence. Is there a potential explanation for this in this cohort?

Author response: Unfortunately, we have no explanation for this. We have added a new Supplement Figure 3A showing the individual levels of sCD163 in patients with or without diabetes (divided into treated and non-treated), stratified by resectability.

  1. The AUC results show that CA 19-9 is quite a strong marker on its own and the addition of sCD163 only marginally improved prediction - not sure whether sCD163 presents any additional benefit in a clinical setting. Also, were these ROC/ AUC measurements also further adjusting for other clinical markers?

Author response: We have updated Figure 2 showing the ROC/AUC curves in the operated and non-operated patients separately. As this is an exploratory biomarker study and we do not have a validation cohort, the ROC/AUC measurements were not adjusted for other clinical markers to avoid overfitting the model.  

  1. The follow-up duration of the study allows for some interesting associations to be investigated. Are there any additional samples at later time points to assess whether there are changes in sCD163? 

Author response: In the present study, we did not measure sCD163 during treatment with chemotherapy or follow-up after an operation. However, we have two ongoing studies of patients with metastatic pancreatic cancer treated with palliative chemotherapy or immunotherapy, and we plan to analyze sCD163 before and during treatment in these patients.  We have added at sentence in the Discussion regarding these planned studies.

  1. Being overweight or obese have been associated with increased sCD163 measurements and present increased risk for pancreatic cancer - yet this was not taken into account or presented in this paper. How does adjustment for waist circumference/ BMI affect the findings?

Author response: Unfortunately, we have not measured waist circumference in the patients. Information on BMI has been added to Table 1. We have also added a new Supplementary Figure 3B showing sCD163 in operated and non-operated patients according to low (<18.5), normal (18.5-25), and high BMI (>25).   

  1. Not discussed in the manuscript are the results from Supplementary Figure S2 - I find the sex differences, particularly, in stage 1 quite interesting. How can you explain these findings and were the differences at each stay by sex significantly different?   

Author response: We have added a new Supplementary Table 6 showing the median (range) and p-values between sCD163 in men and females according to stage and sex. Only for females with stage 1 disease, the median is higher than in men. However, these results should be interpreted with caution due to a low number of patients (n = 6). Unfortunately, we cannot explain these findings, and it would require the inclusion of more samples to fully elucidate if there is a difference in the level of sCD163 between men and females.   

Reviewer 2 Report

This research evaluated the diagnostic and prognostic value of sCD163 using one of the largest studies of patients with PDAC. The key protein was a biomarker previously linked with several cancer types (1). Through enzyme-linked immunosorbent assay (ELISA), the level of sCD163 of 255 patients was tested, and an elevation with statistical significance was discovered between PDAC patients and healthy controls. Cachexia and stent were the two clinical characteristics linked with sCD163 level. Further studies found no evidence to support the use of sCD163 as a prognostic biomarker. The diagnostic value of sCD163 was evaluated, and its value was not higher than CA199, which is the traditional biomarker for PDAC. In terms of novelty, The macrophage-associated molecule CD163 has been reported as a prognostic biomarker in different cancer types by An enzyme-linked immunosorbent assay (ELISA) method. Int J Mol Sci. 2020 Aug 18;21(16):5925.

 1.       Although several studies were conducted, few of them provided a positive result, the test conducted in this study failed to add to sCD163’s significance in PDAC patients, which is the most important weakness of this study.

2.       No reasons were given to explain why sCD163 was chosen to be studied. And as a M2 Macrophage marker, no discussion was done to link sCD163 with macrophage behaviors.

3.       In the discussion part, the author failed to discuss the potential linkage between sCD163 and the few clinical characteristics that sCD163 were correlated with.

4.       The method of this study, using ELISA to study sCD163 expression within serum of PDAC patients, is acceptable. But need to be correlated to the expression of tumor tissues such as TMA?

5.       The authors failed to demonstrate sCD163’s importance in PDAC patients, and more studies should be conducted to discover more positive results (e.g., introducing new methods of calculation or new clinical characteristics).

6.       The authors should change the title of this article, as the title should include the indicated finding (e.g., Combining plasma sCD163 with CA19-9 increases the predictiveness of PDAC)

7.       Few explanations were given on why the authors chose to study sCD163 in PDAC, more explanation should be added in the introduction part.

8.       None of the figures in this article suggested satisfactory results, and the authors should look into that and try to construct more positive results (possibly by changing the methods of analysis, such as, the sensitivity and specificity of sCD163, and also compare to the other markers, replacing OS with DFS in the Kaplan-Meier plot).

9.       If the results of sCD163 were unsatisfactory, you could refine your work by adding measurements of biomarkers related to sCD163 (2).

Figure 1:

1.       Replace “Blood donor” with ‘Healthy control.’

2.       Statistical significance was only discovered between Healthy control and PDAC patients. The plasma sCD163 level is not related to stage, as neither of the two stages exhibited a difference with statistical significance.

3.       Since sCD163 is not related to PDAC stage, try reconstructing the boxplot with other tumor characteristics.

Figure 2:

1.       Replace CD163 with sCD163 in the plot.

2.       The content of plot (a) and plot (b) is in included within plot (c). Suggestion was to delete plot (a) and plot (b), in the meantime, enlarge plot (c).

3.       The AUC of sCD163 was an unsatisfactory 0.78; more tests were needed to research its value in PDAC patients.

Figure 3:

1.       Replace CD163 with sCD163 in the plot.

2.       Since statistical significance was not discovered, you can try changing the cut-off value or using Disease Free Survival instead of OS.

Table 1:

1.       Since Cachexia and stent were the only clinical characteristics with statistical significance, more work should be done to link sCD163 with characteristics like weight loss (3).

Table 3:

1.       As the content of Table 3 and Table 4 are similar, Table 3, as it failed to show positive results, should be moved into the supplementary file.

Reference:

1.     Qian S, Zhang H, Dai H, Ma B, Tian F, Jiang P, et al. Is sCD163 a Clinical Significant Prognostic Value in Cancers? A Systematic Review and Meta-Analysis. Front Oncol. 2020;10:585297.

2.     Yu J, Ploner A, Kordes M, Lohr M, Nilsson M, de Maturana MEL, et al. Plasma protein biomarkers for early detection of pancreatic ductal adenocarcinoma. Int J Cancer. 2021;148(8):2048-58.

3.     Dong J, Yu J, Li Z, Gao S, Wang H, Yang S, et al. Serum insulin-like growth factor binding protein 2 levels as biomarker for pancreatic ductal adenocarcinoma-associated malnutrition and muscle wasting. J Cachexia Sarcopenia Muscle. 2021;12(3):704-16.

Author Response

The overall response to reviewer 2: We sincerely thank the reviewer for the positive evaluation of our manuscript. We also thank the reviewer for taking the time to thoroughly read the manuscript and for the constructive critique ultimately improving the manuscript. Below is the point-by-point response to the comments.

Comments from Reviewer 2

This research evaluated the diagnostic and prognostic value of sCD163 using one of the largest studies of patients with PDAC. The key protein was a biomarker previously linked with several cancer types (1). Through enzyme-linked immunosorbent assay (ELISA), the level of sCD163 of 255 patients was tested, and an elevation with statistical significance was discovered between PDAC patients and healthy controls. Cachexia and stent were the two clinical characteristics linked with sCD163 level. Further studies found no evidence to support the use of sCD163 as a prognostic biomarker. The diagnostic value of sCD163 was evaluated, and its value was not higher than CA199, which is the traditional biomarker for PDAC. In terms of novelty, The macrophage-associated molecule CD163 has been reported as a prognostic biomarker in different cancer types by An enzyme-linked immunosorbent assay (ELISA) method. Int J Mol Sci. 2020 Aug 18;21(16):5925.

  1. Although several studies were conducted, few of them provided a positive result, the test conducted in this study failed to add to sCD163’s significance in PDAC patients, which is the most important weakness of this study.

Author response: We think that our result of elevated sCD163 in patients with PDAC is important to report.

  1. No reasons were given to explain why sCD163 was chosen to be studied. And as a M2 Macrophage marker, no discussion was done to link sCD163 with macrophage behaviors.

Author response: We have added new text in the Discussion regarding sCD163 and macrophage behaviors.  

  1. In the discussion part, the author failed to discuss the potential linkage between sCD163 and the few clinical characteristics that sCD163 were correlated with.

Author response: We have added new text in the Discussion regarding the few clinical characteristics sCD163 correlated with.  

  1. The method of this study, using ELISA to study sCD163 expression within serum of PDAC patients, is acceptable. But need to be correlated to the expression of tumor tissues such as TMA?

Author response: Unfortunately, we do not have tissue samples available from this cohort. However, in a new cohort of patients with metastatic pancreatic cancer, we will evaluate the relationship between circulating sCD163 and the CD163 expression in tumor tissue. This has been mentioned in the Discussion.

  1. The authors failed to demonstrate sCD163’s importance in PDAC patients, and more studies should be conducted to discover more positive results (e.g., introducing new methods of calculation or new clinical characteristics).

Author response: We agree that more studies should be performed to evaluate the clinical value of circulating sCD163 in patients with different types of cancer. We have added new clinical information in Table 1, Figure 1, Supplementary Table S5 and Table S6, and Supplementary Figure 3 (A, B and C). 

  1. The authors should change the title of this article, as the title should include the indicated finding (e.g., Combining plasma sCD163 with CA19-9 increases the predictiveness of PDAC).

Author response: We have changed the title as suggested.

  1. Few explanations were given on why the authors chose to study sCD163 in PDAC, more explanation should be added in the introduction part.

Author response: We have included text in the Introduction on the selection of investigating sCD163 in our cohort of patients with pancreatic cancer.

  1. None of the figures in this article suggested satisfactory results, and the authors should look into that and try to construct more positive results (possibly by changing the methods of analysis, such as, the sensitivity and specificity of sCD163, and also compare to the other markers, replacing OS with DFS in the Kaplan-Meier plot).

Author response: In our cohort, we have not followed the patients with CT scans routinely after an operation In accordance with the Danish guideless after resection of PDAC). The patient had only a CT-scan if they had symptoms of recurrence. We therefore do not think that DFS will be optimal to test in our cohort.

  1. If the results of sCD163 were unsatisfactory, you could refine your work by adding measurements of biomarkers related to sCD163.

Author response: In Table 2 we have shown the correlation between sCD163 and inflammatory biomarkers (C-reactive protein, IL-6, and YKL-40) and routine blood samples (bilirubin, ALAT, thrombocytes, leucocytes, and neutrophils).   

Figure 1:

Replace “Blood donor” with ‘Healthy control.’

Author Response: This has been changed.

Statistical significance was only discovered between Healthy control and PDAC patients. The plasma sCD163 level is not related to stage, as neither of the two stages exhibited a difference with statistical significance.

Since sCD163 is not related to PDAC stage, try reconstructing the boxplot with other tumor characteristics.

Author response: We have added a new Figure 1B (tumor size in the operated patients) and a new Figure 1C (tumor location in the operated patients).

 Figure 2:

  1. Replace CD163 with sCD163 in the plot.

Author Response: This has been changed.

  1. The content of plot (a) and plot (b) is in included within plot (c). Suggestion was to delete plot (a) and plot (b), in the meantime, enlarge plot (c).

Author Response: Figure 2 has been updated. See the next comment and Reviewer 1, comment 4.

  1. The AUC of sCD163 was an unsatisfactory 0.78; more tests were needed to research its value in PDAC patients.

Author Response: We have added a new Figure 2B (only operated patients) and a new Figure 2C (only patients with locally advanced and metastatic pancreatic cancer).

Figure 3:

  1. Replace CD163 with sCD163 in the plot.

Author Response: This has been changed.

  1. Since statistical significance was not discovered, you can try changing the cut-off value or using Disease Free Survival instead of OS.

Author Response: We have no data on DFS.

Table 1:

Since Cachexia and stent were the only clinical characteristics with statistical significance, more work should be done to link sCD163 with characteristics like weight loss (3).

Author Response: We have added BMI and cachexia to a new Supplementary Figure 3 and divided the patients into resectability. Furthermore, BMI low (<18.5), normal (18.5-25), and high BMI (>25) has been added to Table 1.

Table 3:

As the content of Table 3 and Table 4 are similar, Table 3, as it failed to show positive results, should be moved into the supplementary file.

Author Response: We prefer to have these results in the main manuscript although they are negative.

Round 2

Reviewer 1 Report

The authors have addressed my questions and concerns regarding this paper. No additional comments.

Reviewer 2 Report

Although several studies were conducted, few of them provided a positive result, the test conducted in this study failed to add to sCD163’s significance in PDAC patients, which is the most important weakness of this study.

Have not given sufficient evidences?